# Impact of multiple waves of COVID-19 on healthcare networks in the United States

**Emad M. Hassan**, **Hussam N. Mahmoud** *

Department of Civil and Environmental Engineering, Colorado State University, Fort Collins, CO, United States of America

* Hussam.Mahmoud@colostate.edu

## Abstract

The risk of overwhelming hospitals from multiple waves of COVID-19 is yet to be quantified. Here, we investigate the impact of different scenarios of releasing strong measures implemented around the U.S. on COVID-19 hospitalized cases and the risk of overwhelming the hospitals while considering resources at the county level. We show that multiple waves might cause an unprecedented impact on the hospitals if an increasing number of the population becomes susceptible and/or if the various protective measures are discontinued. Furthermore, we explore the ability of different mitigation strategies in providing considerable relief to hospitals. The results can help planners, policymakers, and state officials decide on additional resources required and when to return to normalcy.

## Introduction

The COVID-19 pandemic has, to date, shown a devastating impact on our society, with more than 49 million confirmed cases and total fatalities exceeding 1,249,000 (as of 6 November 2020) [1]. Global efforts such as early lockdown and travel bans [2, 3] have successfully slowed the virus's spread, as evident by the reduction in the number of people infected and deceased and the subsequent relief of demand on hospitals [4]. Despite their effectiveness, these strong measures led to many dire consequences, including economic [5, 6] and mental health crises [7, 8]. Driven by these challenges, different countries utilized various criteria and strategies to ease previously applied measures and gradually return to normalcy [9]. Relaxing the protective measures included, for example, reopening businesses, schools, restaurants, and recreational facilities [10]. Among other factors, the Fall and Winter seasons are making the situation worse by pushing people back inside closed spaces where the virus has a much easier time spreading than the outdoors [11]. Nevertheless, reopening communities comes with the risk of triggering multiple pandemic waves, which is typically more aggressive than the first wave and could potentially cause devastating social and economic consequences [12].

The concern over multiple waves or spikes of the pandemic has been the subject of various national and international debates. Infected cases from second and third waves of COVID-19 have already been recorded in North America, Europe, and Asia, particularly after easing lockdown orders and allowing indoor activities [1]. The U.S., which has the highest number of

**Data Availability Statement:** All relevant data are within the manuscript and its Supporting Information files.

**Funding:** This study was funded by the cooperative agreement 70NANB15H044 between the National

Institute of Standards and Technology (NIST) and Colorado State University. The content expressed in this paper are the views of the authors and do not necessarily represent the opinions or views of NIST or the U.S Department of Commerce. The George T. Abell Professorship discretionary funding provided support in the form of salary for author HNM.

**Competing interests:** No authors have competing interests.

confirmed COVID-19 cases [1] (see **Fig 1A**), has recently experienced a sudden increase in the number of daily confirmed cases in many states during the last week of October and the first week of November 2020. Common among these states is that they removed the stay-at-home order, reopened schools, and allowed indoor activities, causing a large jump in the confirmed daily cases (see **Fig 1B**). A spike in the daily confirmed cases can be observed in more than 42% of the U.S. counties (the total number of counties is 3,143), as shown in **Fig 1C**. These spikes are forming what appears to be a massive third wave that is expected to have a total number of infected cases exceeding the first and second waves, requiring more protective measures and, in some cases, the need to return to complete lockdown. Understanding the different possibilities and consequences of early easing or removing the previously imposed protective measures is critical for devising effective planning and mitigation policies for reducing social and economic consequences.

Hospitals, among other emergency services, are the frontline in the fight against the COVID-19 pandemic. However, since the start of the pandemic, hospital facilities have experienced tremendous strain brought by the demand exceeding their capacity, forcing them to make hard choices between those who can and cannot receive treatments [13, 14]. The hospitalization services needed for COVID-19 cases are based on case criticality. They can be classified into those needing regular beds (inpatient), intensive care unit (ICU) beds, and ICU beds with mechanical ventilators with the hospitalization services and length of stay being a function of the patient's age [15–17]. Failing to provide adequate and appropriate hospitalization services to those infected can increase the fatality rates, especially for critical cases. In the U.S., the total number of hospitals is 6,630 (see **Fig 1D**) and includes about 961,092 licensed beds, 92,513 ICU beds [18], and 62,000 ICU beds with fully-featured mechanical ventilators [19]. The number of unoccupied beds per county (see **S1 Fig in S1 File**), which can be calculated using the data pertaining to the total number of licensed beds and utilization rates for these beds [18, 20], shows disparities in the distribution of the hospitals where many U.S. counties, have no beds for any patients including COVID-19-related patients [21] (see Material and Methods). This is simply because no hospitals exist in these counties. These disparities can have a devastating impact on the hospitals' outcomes [22] especially for vulnerable populations (aged +60), as shown in **S2 Fig in S1 File**, where no one would have access to neither inpatient nor ICU beds in their counties, as shown in **Fig 1E and 1F**.

In this study, we show the impact of multiple waves of COVID-19 on healthcare networks in the U.S. after states reopening during the Fall and Winter seasons. Here, we define healthcare networks as hospital facilities with licensed beds. We perform a disease transmission analysis at the county level to estimate the expected number of hospitalization cases. We further compare the different estimated number of hospitalized cases with the available hospitalization resources to highlight the impact of waves with different magnitudes on patients' access to medical services. We test different state reopening scenarios along with various percentages of susceptible cases and protection rates to assess each scenario's impact on the expected number of cases needing hospitalization services and the number of counties facing a surge in patients beyond their beds capacity. We also investigate the effectiveness of various mitigation strategies, including enforcing protective measures for longer periods, applying states' lockdown at different time scales, and increasing hospital capacity, on enhancing the hospitals' abilities to provide services for the infected patients. Furthermore, we provide an estimate of the required numbers of inpatient and ICU beds as well as ICU beds with mechanical ventilators for different states' reopening scenarios. The cartographic boundary of the U.S. states and counties were obtained from the United States Census Bureau [23].

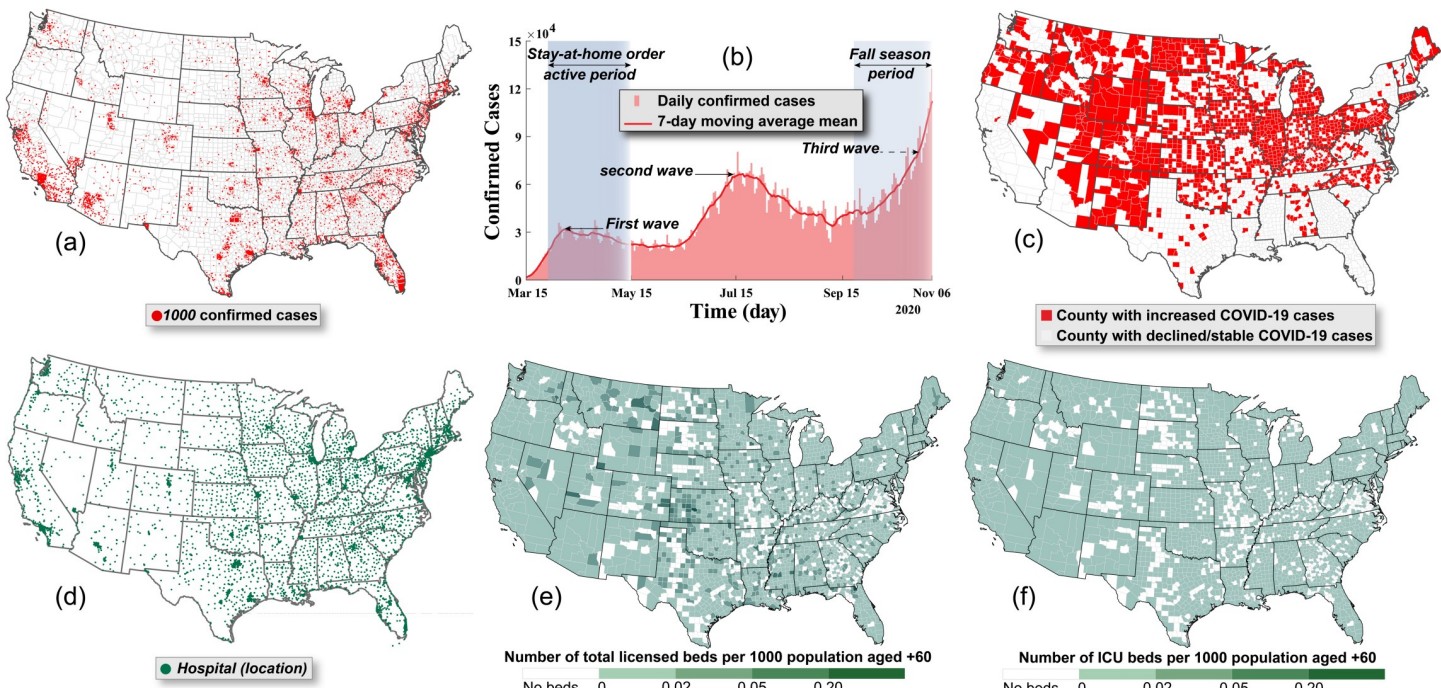

**Fig 1. a**) Distribution of the total number of COVID-19 confirmed cases as of 6 November 2020 in the U.S. [24], **b**) the daily number of confirmed cases in the U.S. with the moving seven-day average means from the initial outbreak to 6 November 2020 [24], **c**) the U.S. counties recording their highest daily number of confirmed cases in the last week of October and the first week of November 2020 (we excluded the counties with mean daily cases less than 5), **d**) the location of U.S. hospitals with licensed staffed beds that can be used to treat COVID-19 patients [25], **e**) the number of licensed beds per 1,000 population aged +60, and **f**) the number of ICU beds per 1000 population aged +60 [20, 25, 26].

## Results

This section discusses the impact of multiple COVID-19 waves on hospitals in the U.S. Some of the results in this section, for certain variables, are presented as the ratio between the calculated value at a given time and that of the peak of the second wave. With a total of 10,067,513 confirmed cases, the U.S. has the highest number of active cases (See Material and Methods for the definition of active cases, *A*) in the world as of 6 November 2020 [1]. This number of cases is expected to increase during the coming weeks, as shown in **Fig 2A**, if the same protection rates and measures, which were applied before the Fall season, are continued as is. Based on a modified SEIR disease transmission model (see Material and Methods), the disease spread prediction is calibrated to data collected until 6 November 2020 and is referred to as the basic case. This basic case does not consider the relaxation of protective measures that some states such as Florida and Utah applied during the Fall season [27]. **Fig 2A** shows the prediction for the aggregated active cases, *A*, in each county, compared with two different datasets [1, 28]. Due to the current limitations and lack of consistency between the reported recovery data [29], we used the average recovery data published by Worldometers [1] and Johns Hopkins [28]. Assuming no easing or relaxing of measures that were applied before Fall season, the active cases' peak is expected to take place in early January of 2021, reaching more than 4.6 million, as shown in **Fig 2A**. Distribution of the COVID-19 cases that need hospitalization services during the peak of the first, second, and third waves are displayed in **S3 Fig in S1 File**, showing, unlike the first wave, the second and third waves to substantially increase the number of hospitalized cases in most counties located in the Mid-America region. Considering the uncertainty associated with the ratio of patients from the active cases requiring hospitalization

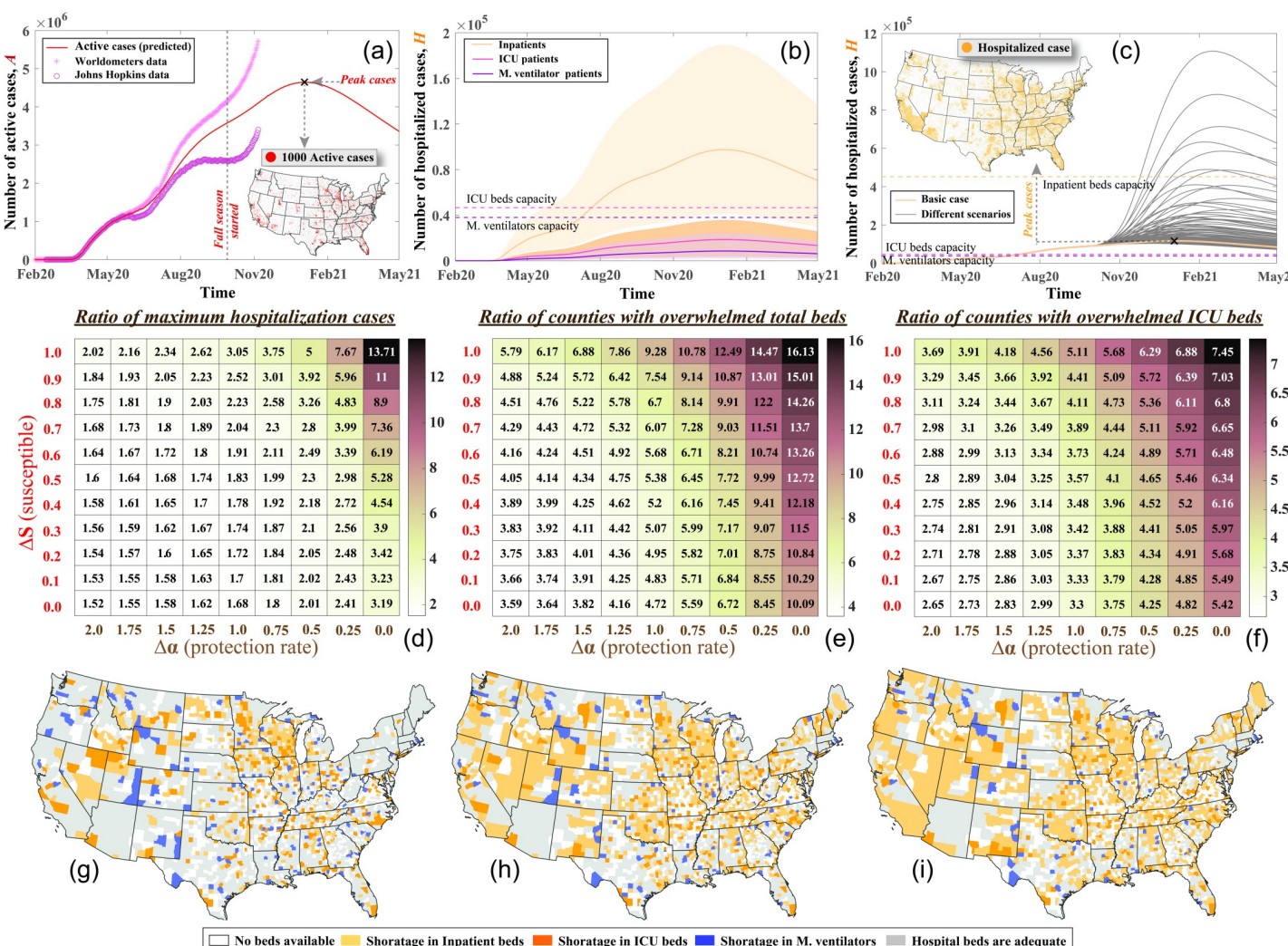

**Fig 2. a)** Fitting of the first and second waves of the active COVID-19 cases in the U.S. compared with the data collected from Worldometers [1] and Johns Hopkins published data [28] sets as well as the predicted active cases for the third wave using the basic case assumptions, **b)** expected number of the hospitalized cases including inpatients, ICU admitted patients, and patients requiring ICUs with mechanical ventilators compared with the available capacity of the U.S. beds in first two categories (the shaded area is the envelope for the 2.5 and 97.5 percentiles for each bed category). **c)** the number of hospitalized cases for different state reopening scenarios compared with the basic case and the distribution of hospitalization cases during the peak of the basic case. The impact of different state reopening scenarios on the ratio of **d)** the maximum number of cases needing hospitalization in the U.S. to the peak hospitalization during the second wave, **e)** the total number of counties with expected hospital demand exceeding the capacity to the maximum number of counties with overwhelmed hospital beds during the second wave, and **f)** the total number of counties with expected ICU demand exceeding the capacity to the maximum number of counties with overwhelmed ICU beds during the second wave. Distribution of counties expected to be overwhelmed with COVID-19 patients for the **g)** basic case, **h)** fully susceptible population with 50% reduction in protection rate, and **i)** fully susceptible population with no protection.

services, the expected number of inpatient and ICU admissions, as well as patients in ICU requiring mechanical ventilators, is shown in **Fig 2B**. This expected demand for the hospitals is compared with the number of unoccupied staffed beds capacity in the U.S. The analysis shows that the U.S. hospitals can handle the maximum demand from COVID-19 cases resulting from the basic case scenario; however, during the peak of the third wave, a shortage of hospital beds and mechanical ventilators is expected in Mid and Southern states as shown in **S3 Fig in S1 File**.

In addition to the basic case, we investigate the impact of eliminating protective measures on increasing the disease's spread. Eliminating the protective measures includes reopening

more schools and workplaces, allowing indoor activities, easing the mask mandatory and social distancing orders, and restoring mobility rates. In this analysis, we use a modified *SEIR* model in which we adjust the percentage of susceptible cases (*S*) and protection rate ($\alpha$) with the time to simulate the increase in mobility and release of protective measures (see Material and Methods) after the deactivation of stay-at-home orders, which resulted sequentially in increasing mobility, easing the mandatory of wearing a mask, reopening schools and workplaces at each state (see **S1 Table in S1 File**). The percent increase in *S* defines the percentage of the population returning to their normal daily routine. An increased $\alpha$ ($\Delta\alpha > 1$) represents more restrictions while a reduced $\alpha$ ($\Delta\alpha < 1$) indicates less restrictive measures such as easing social distancing and not requiring face masks as well as delaying the next stay-at-home order and states lockdown. The case of $\alpha$ equals zero, denotes that no additional restrictions will be applied or lifted. The results show a significant increase in hospitalized cases due to the elimination of protective measures compared with the basic case (see **Fig 2C**). The ratios between the peak of the cases that need hospitalization during each scenario and the peak of the second wave are indicated in **Fig 2D**. The figure shows that enhancing the protective measures (increasing $\alpha$) can reduce the number of hospitalized cases up to 12.8% compared with the basic case but reducing these measures while allowing all population to return to normalcy (i.e., the change in susceptible cases ($\Delta S$) is one), can be catastrophic and could result in hospitalization of 13.7 times that of the peak of the second wave, which increases the demand for hospitals in many counties beyond their capacity, as shown in **Fig 2E and 2F**. These figures show that the change in protection rate ($\Delta\alpha$), which is measured by the ratio of change in protective measures (listed in **S1 Table in S1 File**), is more significant for the spread of the disease than the change in susceptible cases ($\Delta S$), which is measured by the percentage of the population returned to normalcy. In addition, freezing all protective measures ($\Delta\alpha = 0$) while not changing the susceptible cases ($\Delta S = 0$) increases the number of hospitalized cases more than three times that of the second wave. Therefore, maintaining the protective measures is critical in reducing the number of cases and preventing the overwhelming of hospital facilities. More details about the expected number of cases needing inpatient beds, ICU beds, and ICU beds with mechanical ventilators compared with available beds in each state can be found in **S4–S6 Figs in S1 File** for the three scenarios discussed in the following section.

We also identify counties in the U.S. with expected hospitals demand exceeding the county's unoccupied licensed bed capacity, as shown in **Fig 2G–2I** for the basic case, fully susceptible population and 50% protection rate, and fully susceptible population and no protection, respectively. In addition to the counties with no staffed beds for any of the three considered bed types (758 counties), the number of counties that might experience a shortage in inpatient beds is 323 {$2.5p^{th} = 770$, $97.5p^{th} = 60$}, the ICU beds is 586 {$2.5p^{th} = 1117$, $97.5p^{th} = 18$}, and the mechanical ventilators is 526 {$2.5p^{th} = 1260$, $97.5p^{th} = 94$} for the basic case. Most of these counties are in the Mid-America region. The number of patients' overflow in these counties is less than most of the states' bed and ventilator capacity; therefore, the patients can be accommodated by patient transfer to other hospitals within each state. However, for the full susceptible population with 50% reduction in the protection rate of the basic case, the numbers of overwhelmed counties will increase to 894 {$2.5p^{th} = 1302$, $97.5p^{th} = 482$} for the inpatient beds, 1146 {$2.5p^{th} = 1570$, $97.5p^{th} = 322$} for the ICU beds, and 1060 {$2.5p^{th} = 1636$, $97.5p^{th} = 382$} for the mechanical ventilators. For the fully susceptible population and no protection, the number of overwhelmed counties will be 1168 {$2.5p^{th} = 1483$, $97.5p^{th} = 823$} for the inpatient beds, 1365 {$2.5p^{th} = 1687$, $97.5p^{th} = 676$} for the ICU beds, and 1283 {$2.5p^{th} = 1748$, $97.5p^{th} = 689$} for the mechanical ventilators. In these two scenarios, we show that urban counties, despite having a large number of staffed beds, might also be overwhelmed. The distribution for counties expected to be overwhelmed with COVID-19 patients during peak cases,

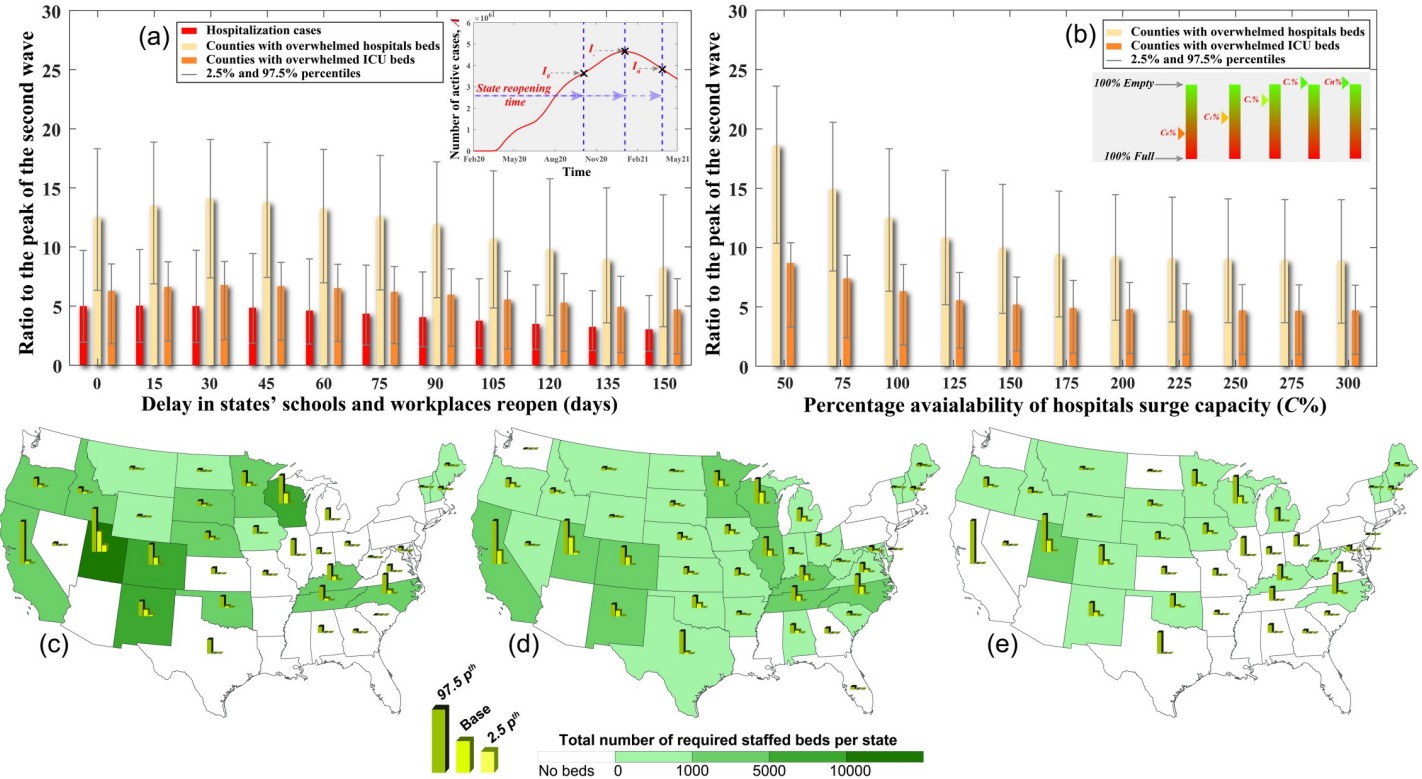

**Fig 3. a)** Impact of delay in states' reopening indoor activities including schools and workplaces on the expected ratio of the peak of the hospitalized cases and the expected number of overwhelmed counties based on the total number of beds and ICU beds to that of the second wave. **b)** Effect of the available surge capacity at each county's hospitals on the expected ratio of overwhelmed counties based on the total number of beds and ICU beds to that of the second wave. Required number of additional staffed beds per state to avoid overwhelming the hospitals based on **c)** inpatient beds, **d)** ICU beds, and **e)** ICU beds with mechanical ventilators. The bar graphs represent the base, 97.5, and 2.5 percentiles, while the color fill in the maps denotes the 50 percentiles.

based on different susceptible cases and protection rates, is shown in **S7 Fig in S1 File** for most of the scenarios summarized in **Fig 2D**. Applying the scenario of easing the protective measures during the Fall and Winter seasons (full susceptible population and 50% protection rate) is also applied to other countries that have different disease spread rates, various protective measures, and dissimilar healthcare system capacities compared with the U.S. as shown in **S8 Fig in S1 File**. We found that easing these protective measures, even with a lower number of active cases, can cause other waves of disease spread that exceed the healthcare system capacity in some countries.

## Discussion

This section investigates different strategies that might help reduce the consequences of partially eliminating protective measures, while states are reopened during the Fall and Winter seasons. Communities and hospital owners and operators commonly use these strategies to decrease the number of hospitalized cases and/or enhance the hospitals' ability and capacity to treat patients. Here, we use the case of a fully susceptible and 50% reduction in protection rate, which we discussed earlier in the results section. Keeping the schools and workplaces open during the Fall and Winter seasons and without protective and strong measures, especially with a large number of active cases from the second wave, can have devastating impacts, including a sudden increase in the number of infected cases and, in some cases, severe multiple waves of disease spread. Our analysis shows that delaying the states' reopening schools,

workplaces, and other indoor activities as well as maintaining strong mitigation measures can efficiently reduce the number of infected and hospitalized cases and decrease the risk of overwhelming the hospitals with patients, as shown in **Fig 3A**. While early reopening ultimately leads to a third wave with a magnitude of 5.0 {$2.5p^{th}$ = 9.7, $97.5p^{th}$ = 1.9} times the second wave, appropriate timing of the reopening can result in minor waves with a substantial reduction in the peak of the hospitalized cases, which can crucially prevent overwhelming of the hospitals. It can also be noticed from the analysis that the impact of reopening is a function of the number of infectious cases at the reopening stage. However, this is not always the case and, if no further protective measures are applied ($\alpha = 0$), this impact will be minimal. On the other hand, applying additional lockdown of states and increasing the protective measures ($\alpha = 2$), even for a short period, can have a more significant reduction on the number of hospitalization cases and counties with overwhelmed hospital beds as shown in **S9 Fig in S1 File**. A two-week lockdown of states can reduce the peak number of hospitalization cases by about 50% and decrease the total number of counties with expected hospital demand exceeding the capacity by more than 60% compared with the case of no lockdown.

One of the main approaches hospitals use to manage patients' sudden increase is to provide surge capacity [30]. This can be realized by reducing regular patients' hospitalization rates to allow more COVID patients to be admitted. The number of regular patients in each county has been documented since the beginning of the pandemic [31]. In this analysis, we assume no increase in licensed beds in each hospital, but we consider the option of operating the licensed and physically available but unstaffed beds to treat COVID-19 related patients. Increasing the surge capacity approach can considerably reduce the number of overwhelmed counties, as shown in **Fig 3B**; however, providing surge capacity is limited by the number of beds in each facility. When compared to the second wave, providing surge capacity can reduce the ratio of counties with overwhelmed hospitals to 8.9 {$2.5p^{th}$ = 14.0, $97.5p^{th}$ = 3.6} and counties with overwhelmed ICU beds to 4.7 {$2.5p^{th}$ = 6.9, $97.5p^{th}$ = 1.0}. On the other hand, the number of overwhelmed counties can significantly increase if the surge in capacity is limited. In such case, the ratio of counties with overwhelmed hospitals can increase to 18.6 {$2.5p^{th}$ = 23.6, $97.5p^{th}$ = 10.4} and counties with overwhelmed ICU beds to 8.7 {$2.5p^{th}$ = 10.4, $97.5p^{th}$ = 3.3} compared with the second wave.

Another method to increase the capacity of the hospitals is to add additional staffed beds. These additional staffed beds can be added as field hospitals [32] or backup beds at the existing hospitals [30]. In this analysis, we identify the states that will need additional beds and quantify the number of beds that will be required. To quantify the optimal number of staffed beds needed at each U.S. state, we first evaluate the expected maximum number of cases needing hospitalization per state and assume that each state's hospitals can manage to treat patients from overwhelmed counties [33]. However, when all hospitals within a state are overwhelmed (see **S5 Fig in S1 File**), additional support will be required to bridge the staffed beds' demand and capacity gap. **Fig 3C–3E** show the number of staffed beds needed per state based on the base, 2.5, and 97.5 percentiles (see Material and Methods) for the inpatient beds, ICU beds, and ICU beds with mechanical ventilators, respectively. The analysis shows that the states located in Mid-America are more vulnerable to their hospitals being overwhelmed and will need additional beds, and mechanical ventilators if states are fully reopened, and a 50% reduction in protection rate is utilized. The required additional inpatient beds, ICU beds, and ICU beds and mechanical ventilators per state for other different scenarios are shown in **S10 Fig in S1 File**.

## Conclusion

In conclusion, we explored the impact of second and third waves of COVID-19 on hospitals in the U.S. We used a modified *SEIR* model to predict the number of hospitalized cases for each

county considering various state reopening scenarios, including partial or fully reopening, while considering different levels of population protection rates. We identified the counties that might experience overwhelming patients demand that exceeds their hospitals' capacity. We further investigated the impact of different mitigation strategies on the number of cases that need hospitalization, hospital availability, and the number of staffed beds that will be needed to overcome the expected shortage of the staffed beds.

The analysis focused on estimating the cases that need hospitalization while considering the available resources in each county. We assumed the number of recovered cases in each county based on the recovery rates in the U.S. due to data limitations. We evaluated the uncertainty in the hospitalized cases and fitted the disease transmission model to published data to estimate the disease model parameters. However, utilizing more data could lower the level of uncertainties in these estimates. We assumed that the population per county is constant, and we neglected the impact of the relocation between states on disease spread. Furthermore, we used published data to estimate the number of staffed beds per county and the utilization of these beds, and we are not accounting for the additional staffed beds and field hospitals built after the pandemic outbreak in the U.S. We also did not include the effect of the population vaccinated for the virus nor the different virus mutations on the forecasted disease spread.

## Material and methods

### Disease transmission model

Among many other disease transmission models [34], the *SEIR* models are commonly used to model the COVID-19 pandemic [35, 36]. We developed a modified version of the generalized six-states *SEIR* disease transmission model [37] to include ten different states as follow {$S$, $P$, $E$, $I$, $Q$, $T$, $C$, $V$, $R$, $D$}$_t$ to represent the susceptible, insusceptible, exposed, infective, self-quarantined, inpatient admitted, *ICU* admitted, cases on a mechanical ventilator, recovered, and deceased cases, respectively as a function of time, *t*. The additional four cases ($Q$, $T$, $C$, and $V$) represent different types of confirmed cases based on their hospitalization services needs in which $Q$ is for cases with mild or no symptoms, $T$ is for cases needing hospital admission, $C$ is for cases needing ICU, and $V$ is for cases needing mechanical ventilators. These four states can be aggregated to represent the total number of active cases, $A$, which represents all the positive (confirmed) cases with no outcomes (recovered or deceased) yet. $T$, $C$, and $V$ cases together form the total COVID-19 demand on the hospitals, $H$. The following model is constructed for population, $N$, of each county, $i$, in the U.S. The differential equations below are used to determine the total number in each state in county $i$.

$$dS/dt = -\beta\,(SI)/N - \alpha S \tag{1}$$

$$dP/dt = \alpha S \tag{2}$$

$$dE/dt = \beta\,(SI)/N - \gamma E \tag{3}$$

$$dI/dt = \gamma E - \delta I \tag{4}$$

$$dQ/dt = \delta I - \zeta Q - \lambda_Q Q - v_Q Q \tag{5}$$

$$dT/dt = \zeta Q - \eta T - \lambda_H T - v_H T \tag{6}$$

$$dC/dt = \eta T - \kappa C - \lambda_C C - v_C C \tag{7}$$

$$dV/dt = \kappa C - \lambda_V V - \nu_V V \qquad (8)$$

$$dR/dt = \lambda_Q Q + \lambda_H T + \lambda_C C + \lambda_V V \qquad (9)$$

$$dD/dt = \nu_Q Q + \nu_H T + \nu_C C + \nu_V V \qquad (10)$$

Where, $\beta$ is the infection rate, $\alpha$ is the protection rate, $1/\gamma$ is the average incubation period, $1/\delta$ is the average quarantine time, $\zeta$ is the hospitalization rate, $\eta$ is the *ICU* rate, and $\kappa$ is the mechanical ventilator rate. In addition, $\lambda_Q$, $\lambda_H$, $\lambda_C$, and $\lambda_V$ are the recovery rates for self-quarantined, inpatient, ICU, and mechanical ventilator cases, respectively. Moreover, $\nu_Q$, $\nu_H$, $\nu_C$, and $\nu_V$ are the death rate for the self-quarantined, inpatients, and patients in the ICU, and those on a mechanical ventilator, respectively. Furthermore, the basic reproduction number, $R_0$, is the average number of secondary infective cases produced by one infective case in the same county during the infectious period of this case and equals to $\beta/\delta(1-\alpha)^t$. In the utilized model we assume a constant population for each investigated county over the epidemic time, $N$, which satisfies the equilibrium of $N = S+P+E+I+Q+T+C+V+R+D$ at any time $t$.

Using the modified *SEIR* model and including self-quarantined and hospitalized states while accounting for protective measures' impact allows for more reliable fitting and forecasting of the COVID-19 disease spread. Assigning positive values to the protection rate, $\alpha$, simulates different protective measures, including lockdown, social distancing, wearing protective masks, etc. To simulate state reopening, we model different percentages of the population who return to normalcy and become non-protected ($\Delta S$) in the reopened counties by increasing the number of susceptible populations at the time of stay-at-home deactivation and resuming time for schools and businesses (see **S1 Table in S1 File**). While modeling the strengthening or easing of each county's protective measures, such as mask mandatory wearing orders, is realized by changing the protection rate as a ratio of the protection rate for the basic case ($\Delta\alpha$) at the county reopening or schools and businesses resuming date (see **S1 Table in S1 File**). Shifting the $\alpha$ can change the disease spread rate and the basic reproduction number, $R_0$, which can be reduced with time [37] using protective measures [38].

Estimation of the model parameters ($\beta$, $\alpha$, $\gamma$, $\delta$, and $\nu$(s)) is made by fitting the published data for confirmed and deceased [24], while $\lambda$(s) are estimated as a time-dependent parameter from the published US recovery data [1, 28] and assumed to be similar for all the US counties. Initial values for the parameter estimation are assumed based on previous studies [16, 39] and CDC reports [15, 38]. During the state's reopening and resuming of schools and businesses, $\alpha$ is modified based on the investigated scenario. We simulated the $\zeta$ as gamma distribution with 0.025, 6.33, and 0.004 for base, shape, and scale parameters, respectively, while for $\eta$ we used gamma distribution with 0.16, 6.13, and 0.02 for base, shape, and scale parameters, respectively, and for $\kappa$ we utilized beta distribution with 0.46, 5.22, and 3.08 for base, shape, and scale parameters, respectively [16, 39]. These distributions are used to model the uncertainty associated with the number of different hospitalization cases, in which we use Monto-Carlo simulations with 100,000 trials.

## Hospitals model

Modeling the hospitals' capacity in the U.S. is based on the published data for all the U.S. hospitals, including hospital location, the number of licensed/staffed beds and ICU beds, and the utilization ratio of these beds [18, 20]. We aggregate these data to calculate the number of total inpatient and ICU beds in each county. Due to the limited data on the number of mechanical

ventilators per county, we assume that the number of mechanical ventilators per the ICU bed is constant and is based on recently published estimates of ventilators in the U.S. [19]. To simulate the available surge capacity (unoccupied beds) per county, we use the number of licensed staffed beds multiplied by the utilization rates, while considering the potential use of licensed and physically available but unstaffed beds. The capacity calculations are then verified with the data from the CDC dashboard [40]. These beds are used in our analysis for COVID-19 cases that require hospitalization service. COVID-19 patients from counties with no staffed beds are redistributed to unoccupied beds in hospitals within the same state. For the cases where no beds are available in the whole state, the patient is considered untreated. The distribution of patients is realized using a patient-driven model, previously developed by Hassan and Mahmoud [41], to determine the most probable hospital.

## Supporting information

**S1 File.**
(DOCX)

## Author Contributions

**Conceptualization:** Emad M. Hassan, Hussam N. Mahmoud.

**Data curation:** Emad M. Hassan.

**Formal analysis:** Emad M. Hassan.

**Methodology:** Emad M. Hassan, Hussam N. Mahmoud.

**Writing – original draft:** Emad M. Hassan.

**Writing – review & editing:** Hussam N. Mahmoud.

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
