## [Decision Letter · Decision Letter 0]

11 Nov 2020

PONE-D-20-26037

Impact of COVID-19 Second Wave on Healthcare Networks in the United States

PLOS ONE

Dear Dr. Mahmoud,

Thank you for submitting your manuscript to PLOS ONE. After careful consideration, we feel that it has merit but does not fully meet PLOS ONE’s publication criteria as it currently stands. Therefore, we invite you to submit a revised version of the manuscript that addresses the points raised during the review process.

     An updated information on COVID-19 in US; the one referenced in the manuscript dates back in June;(this must include updates on counties used for the model, beds and ICU facilities etc)A rebuttal letter that responds to each point raised by the academic editor and reviewer(s). You should upload this letter as a separate file labeled 'Response to Reviewers'.A marked-up copy of your manuscript that highlights changes made to the original version. You should upload this as a separate file labeled 'Revised Manuscript with Track Changes'.An unmarked version of your revised paper without tracked changes. You should upload this as a separate file labeled 'Manuscript'.

We look forward to receiving your revised manuscript.

Kind regards,

Rashid Ansumana

Academic Editor

PLOS ONE

Additional Editor Comments:

In additional to reviewer comments, authors are advised to update the information used for the model to reflect the current state of affairs in the counties used for the model.

Journal Requirements:

2. Please place Materials and Methods and Results in separate sections for the reader's ease and in accordance with the journal's submission guidelines.

" Funding for this study was in part provided by the cooperative agreement 70NANB15H044 between the National Institute of Standards and Technology (NIST) and Colorado State University. "

4. We note that Figures 1, 2, 3, S1, S2, S6 and S7 in your submission contain map images which may be copyrighted. All PLOS content is published under the Creative Commons Attribution License (CC BY 4.0), which means that the manuscript, images, and Supporting Information files will be freely available online, and any third party is permitted to access, download, copy, distribute, and use these materials in any way, even commercially, with proper attribution. For these reasons, we cannot publish previously copyrighted maps or satellite images created using proprietary data, such as Google software (Google Maps, Street View, and Earth). For more information, see our copyright guidelines: http://journals.plos.org/plosone/s/licenses-and-copyright.

4.1.    You may seek permission from the original copyright holder of Figures 1, 2, 3, S1, S2, S6 and S7 to publish the content specifically under the CC BY 4.0 license. 

4.2.    If you are unable to obtain permission from the original copyright holder to publish these figures under the CC BY 4.0 license or if the copyright holder’s requirements are incompatible with the CC BY 4.0 license, please either i) remove the figure or ii) supply a replacement figure that complies with the CC BY 4.0 license. Please check copyright information on all replacement figures and update the figure caption with source information. If applicable, please specify in the figure caption text when a figure is similar but not identical to the original image and is therefore for illustrative purposes only.

Reviewers' comments:

Reviewer's Responses to Questions

**Comments to the Author**

1. Is the manuscript technically sound, and do the data support the conclusions?

Reviewer #1: Yes

2. Has the statistical analysis been performed appropriately and rigorously? 

Reviewer #1: Yes

3. Have the authors made all data underlying the findings in their manuscript fully available?

Reviewer #1: Yes

4. Is the manuscript presented in an intelligible fashion and written in standard English?

Reviewer #1: Yes

5. Review Comments to the Author

Reviewer #1: Dear authors,

Thank you for your efforts in this manuscript which is really very interesting. I have few comments to be considered

1- in the result section, could you emphasize more on the findings based on the models as you reported that you did the analysis without commenting on it

2- Could you compare between the situation in the US and other nearby countries with similar healthcare systems

3- Could you discuss the different suggested protocols to handle the situation, specially because relaxing measures are fact now and countries should try to deal with the consequences

4- Could you present a model of country in which they managed to overcome the burden on healthcare system while curfew measures are not so strict

6. PLOS authors have the option to publish the peer review history of their article (what does this mean?). If published, this will include your full peer review and any attached files.

Reviewer #1: No

---

## [Author Response · Author response to Decision Letter 0]

19 Dec 2020

Impact of Multiple Waves of COVID-19 on Healthcare Networks in the United States

RESPONSE TO EDITOR GENERAL COMMENTS:

Thank you for submitting your manuscript to PLOS ONE. After careful consideration, we feel that it has merit but does not fully meet PLOS ONE’s publication criteria as it currently stands. Therefore, we invite you to submit a revised version of the manuscript that addresses the points raised during the review process. In additional to reviewer comments, authors are advised to update the information used for the model to reflect the current state of affairs in the counties used for the model.

Authors Response: Thank you for the overall reflection and feedback on the manuscript. We modified the paper to address the editor’s comments and all comments provided by the reviewer. The modifications include a) updating the data used for the disease transmission model, healthcare facilities, and the licensed and staffed beds in each U.S. county, and b) reflecting more on our findings and including additional mitigation strategies that represent the current situation.

Please find below our detailed response to the reviewer’s comments. Unless otherwise specified, the page numbers in the responses refer to the pages in the revised manuscript with highlighted changes. Changes made to address the comments appear in the manuscript in blue font.

 

RESPONSE TO JOURNAL REQUIREMENT COMMENTS:

 Authors Response: Thank you we followed PLOS ONE’s style requirement.

2. Please place Materials and Methods and Results in separate sections for the reader's ease and in accordance with the journal's submission guidelines.

Authors Response: Thank you we placed the Materials and Methods and Results in separate sections.

" Funding for this study was in part provided by the cooperative agreement 70NANB15H044 between the National Institute of Standards and Technology (NIST) and Colorado State University. "

Authors Response: Thank you. We modified the acknowledgment section to include all funding sources. We also included the funding statement in the attached cover letter.

4. We note that Figures 1, 2, 3, S1, S2, S6 and S7 in your submission contain map images which may be copyrighted. All PLOS content is published under the Creative Commons Attribution License (CC BY 4.0), which means that the manuscript, images, and Supporting Information files will be freely available online, and any third party is permitted to access, download, copy, distribute, and use these materials in any way, even commercially, with proper attribution. For these reasons, we cannot publish previously copyrighted maps or satellite images created using proprietary data, such as Google software (Google Maps, Street View, and Earth). For more information, see our copyright guidelines: http://journals.plos.org/plosone/s/licenses-and-copyright.

You may seek permission from the original copyright holder of Figures 1, 2, 3, S1, S2, S6 and S7 to publish the content specifically under the CC BY 4.0 license. 

Authors Response: Thank you for the comment. The actual map of the U.S. including the state and county boundaries were obtained from the U.S. Census Bureau. The respective file (shapefile) contains the following note about “Use limitations”: “These products are free to use in a product or publication, however, the acknowledgment must be given to the U.S. Census Bureau as the source.” As such, there is no need to obtain permission for the actual map since we acknowledged the U.S. Census Bureau as the source in the revised manuscript. [please see page 4].

The data that is populating the maps in Figure 1 (a), (b), and (c) were downloaded from USAFACTS webpage and can be used and downloaded as mentioned by the webpage: “The underlying data is available for download” and we citied the data source in the figure caption [please see page 4]. As such, there is no need to obtain permission from USAFACTS to use this data.

The healthcare system data that are populating the maps in figures 1 (d), (e), and (f), as well as figure S1 in the modified documents were downloaded from Definitive Healthcare. The license for these data is customized and can be used for non-commercial purposes as mentioned in the license agreement attached to the data source: “Any reference to this data for non-commercial purposes must identify Definitive Healthcare as the source of this information by citing the information with the following attribution: https://www.definitivehc.com/.”. As such, there is no need to obtain permission for the use of these data since we acknowledged the Definitive Healthcare as shown in the caption for figure 1.

The population data that are populating the maps in Figure 1 (e) and (f), as well as Figure S2, in the modified documents were downloaded from the U.S. Census Bureau. The data description section contains the following note about “Use limitations”: “These products are free to use in a product or publication, however, the acknowledgment must be given to the U.S. Census Bureau as the source.” As such, there is no need to obtain permission for the actual map since we acknowledged the U.S. Census Bureau as the source in the revised manuscript as shown below. [please see page 4]

The data used to develop Figure 2 and 3, as well as figures S3, S7, and S10 in the modified documents, were created by the authors by processing public (raw) data that already cited in the manuscript and include a) U.S. Census Bureau, b) USAFACTS, c) and Definitive Healthcare. As such, no permission is needed as well. The specifics of how these data were processed are mentioned in the manuscript. [please see Results, Discussions, and Material and Methods sectiona]

The respective data and maps utilized to create the figures are public and are available in the links below:

1. Bureau UC. Cartographic boundary files—shapefile:

https://www.census.gov/geographies/mapping-files/time-series/geo/carto-boundary-file.html

2. USAFACTS. Coronavirus locations: COVID-19 map by county and state:

https://usafacts.org/visualizations/coronavirus-covid-19-spread-map/

3. Definitive Healthcare. USA Hospital Beds:

https://coronavirus-resources.esri.com/datasets/definitivehc::definitive-healthcare-usa-hospital-beds?geometry=97.382%2C-16.820%2C-122.344%2C72.123&selectedAttribute=NUM_ICU_BEDS

4. U.S. Census Bureau. American Community Survey 1-year estimates:

http://censusreporter.org/profiles/01000US-united-states/

 

RESPONSE TO REVIEWER 1:

General Comments: 

Dear authors, Thank you for your efforts in this manuscript which is really very interesting. I have few comments to be considered.

Authors Response: We sincerely appreciate all the time the reviewer spent in reading the manuscript and providing extensive and thoughtful comments. We modified the manuscript to address all comments and suggestions, as noted below, which has enabled us to improve the quality of the paper significantly. 

Specific Comments:

1. in the result section, could you emphasize more on the findings based on the models as you reported that you did the analysis without commenting on it 

Authors Response: Thank you for your comments. We modified the results section to reflect more on the main findings of the study by discussing the details of the distribution of the COVID-19 cases during the second and third waves (please note that we also added the results for the third wave as per the suggestion by the editor). [please see pages 4, 5, and 6]

2. Could you compare between the situation in the US and other nearby countries with similar healthcare systems 

Authors Response: We appreciate this comment as well and wish to thank the reviewer for his/her suggestion. To address this comment, we introduced a comparison between the disease spread in the U.S. and other countries including Germany, China, and India. We added this comparison in the Supporting Information document and we commented on this comparison in the manuscript. [please see page 6 in the main paper as well as page 10 and Figure S8 in Supporting Information]

3. Could you discuss the different suggested protocols to handle the situation, specially because relaxing measures are fact now and countries should try to deal with the consequences

Authors Response: Thank you for raising the mentioned issue. We have analyzed and reported, in the original submission, the consequences of the case where no protective measures are applied as shown in Fig. 2. To address the reviewer's comment we modified the manuscript to discuss the different suggestions that might be implemented while relaxing the protective measures. [please see pages 7 and 8 in the main paper as well as page 12 and Figure S9 in the Supporting Information]

4. Could you present a model of country in which they managed to overcome the burden on healthcare system while curfew measures are not so strict

Authors Response: We would like to thank the reviewer for his comment. In our response to the first comment of the reviewer, we included countries that applied various levels of protective measures including complete and partial lockdown, mandating social distance and face cover, and tracking possible exposed cases. To address this comment, we discussed in the modified manuscript the impact of the various protective measures applied by countries including Germany, China, and India on the expected number of hospitalization cases and we reflect on how these countries managed their healthcare system during the pandemic. [please see page 6 in the main paper as well as page 10 and Figure S8 in Supporting Information]

---

## [Decision Letter · Decision Letter 1]

8 Feb 2021

Impact of Multiple Waves of COVID-19 on Healthcare Networks in the United States

PONE-D-20-26037R1

Dear Dr. Mahmoud,

We’re pleased to inform you that your manuscript has been judged scientifically suitable for publication and will be formally accepted for publication once it meets all outstanding technical requirements.

Kind regards,

Itamar Ashkenazi

Academic Editor

PLOS ONE

Additional Editor Comments (optional):

Reviewers' comments:

Reviewer's Responses to Questions

**Comments to the Author**

1. If the authors have adequately addressed your comments raised in a previous round of review and you feel that this manuscript is now acceptable for publication, you may indicate that here to bypass the “Comments to the Author” section, enter your conflict of interest statement in the “Confidential to Editor” section, and submit your "Accept" recommendation.

Reviewer #1: All comments have been addressed

2. Is the manuscript technically sound, and do the data support the conclusions?

Reviewer #1: Yes

3. Has the statistical analysis been performed appropriately and rigorously? 

Reviewer #1: Yes

4. Have the authors made all data underlying the findings in their manuscript fully available?

Reviewer #1: Yes

5. Is the manuscript presented in an intelligible fashion and written in standard English?

Reviewer #1: Yes

6. Review Comments to the Author

Reviewer #1: Thank you for your efforts in replying all the comments in an intelligible and attractive way. The new copy is really interesting

7. PLOS authors have the option to publish the peer review history of their article (what does this mean?). If published, this will include your full peer review and any attached files.

Reviewer #1: No

---

## [Editor Report · Acceptance letter]

22 Feb 2021

PONE-D-20-26037R1 

Impact of Multiple Waves of COVID-19 on Healthcare Networks in the United States 

Dear Dr. Mahmoud:

I'm pleased to inform you that your manuscript has been deemed suitable for publication in PLOS ONE. Congratulations! Your manuscript is now with our production department. 

Kind regards, 

on behalf of

Dr. Itamar Ashkenazi 

Academic Editor

PLOS ONE